# Genome-Scale Metabolic Modelling of Lifestyle Changes in *Rhizobium leguminosarum*

Carolin C. M. Schulte,[a,b] Vinoy K. Ramachandran,[a] Antonis Papachristodoulou,[b] Philip S. Poole[a]

aDepartment of Plant Sciences, University of Oxford, Oxford, UK
bDepartment of Engineering Science, University of Oxford, Oxford, UK

**ABSTRACT** Biological nitrogen fixation in rhizobium-legume symbioses is of major importance for sustainable agricultural practices. To establish a mutualistic relationship with their plant host, rhizobia transition from free-living bacteria in soil to growth down infection threads inside plant roots and finally differentiate into nitrogen-fixing bacteroids. We reconstructed a genome-scale metabolic model for *Rhizobium leguminosarum* and integrated the model with transcriptome, proteome, metabolome, and gene essentiality data to investigate nutrient uptake and metabolic fluxes characteristic of these different lifestyles. Synthesis of leucine, polyphosphate, and AICAR is predicted to be important in the rhizosphere, while *myo*-inositol catabolism is active in undifferentiated nodule bacteria in agreement with experimental evidence. The model indicates that bacteroids utilize xylose and glycolate in addition to dicarboxylates, which could explain previously described gene expression patterns. Histidine is predicted to be actively synthesized in bacteroids, consistent with transcriptome and proteome data for several rhizobial species. These results provide the basis for targeted experimental investigation of metabolic processes specific to the different stages of the rhizobium-legume symbioses.

**IMPORTANCE** Rhizobia are soil bacteria that induce nodule formation on plant roots and differentiate into nitrogen-fixing bacteroids. A detailed understanding of this complex symbiosis is essential for advancing ongoing efforts to engineer novel symbioses with cereal crops for sustainable agriculture. Here, we reconstruct and validate a genome-scale metabolic model for *Rhizobium leguminosarum* bv. *viciae* 3841. By integrating the model with various experimental data sets specific to different stages of symbiosis formation, we elucidate the metabolic characteristics of rhizosphere bacteria, undifferentiated bacteria inside root nodules, and nitrogen-fixing bacteroids. Our model predicts metabolic flux patterns for these three distinct lifestyles, thus providing a framework for the interpretation of genome-scale experimental data sets and identifying targets for future experimental studies.

**KEYWORDS** Rhizobium leguminosarum, metabolic modeling, rhizosphere-inhabiting microbes, symbiosis

Nitrogen is commonly the main limiting nutrient in agriculture because plants are unable to assimilate atmospheric $N_2$ (1). Some legumes, such as peas, beans, and lentils, circumvent this problem by entering into complex symbiotic relationships with soil bacteria called rhizobia. Legumes secrete signaling molecules (flavonoids) that are recognized by compatible rhizobia, which produce their own signaling molecules (Nod factors) in response. As a result of this signal exchange, rhizobia are typically entrapped by root hairs and grow down so-called infection threads until they are endocytosed by plant cells in the developing nodule. The bacteria then undergo further cell division and eventually differentiate into bacteroids converting atmospheric $N_2$ into ammonia, which is secreted to the plant host in exchange for carbon sources, mainly dicarboxylates (2–4).

Address correspondence to Antonis Papachristodoulou, antonis@eng.ox.ac.uk, or Philip S. Poole, philip.poole@plants.ox.ac.uk.
The authors declare no conflict of interest.

Symbiosis formation is a multi-stage process, requiring distinct metabolic capabilities at each stage. The ability of rhizobia to adapt to various environmental conditions is reflected in their large genomes, which often comprise several replicons (5–7), and in the importance of different genomic regions for each lifestyle (8, 9). While significant research efforts have focused on understanding bacteroid metabolism in rhizobium-legume symbioses, several recent studies have begun to unravel the plant-bacteria interactions preceding the formation of differentiated nitrogen-fixing bacteroids. For example, transcriptomic changes in response to root exudates of different plants have been investigated (10, 11) and biosensors have been developed to elucidate nutrient availability in the rhizosphere (12). Importantly, a study using transposon-based insertion sequencing (INSeq) assessed gene essentiality in *Rhizobium leguminosarum* for rhizosphere bacteria, root-attached bacteria, undifferentiated nodule bacteria, and nitrogen-fixing bacteroids (13). It was found that 603 genetic regions were essential for a successful transition from free-living bacteria to bacteroids, highlighting the complexity of development during formation of a successful symbiosis. Understanding the metabolic features at different stages of symbiosis is required for developing effective rhizobial inocula for agricultural applications. Rhizobia that efficiently fix nitrogen are not necessarily adapted to persistence in the rhizosphere as well as nodulating a plant host in the presence of genetically different bacterial strains, a characteristic described as competitiveness (13–15). Knowledge of the nutrient exchanges between plants and rhizosphere bacteria is thus required for the design of microbial inocula that are competitive and stably persist when applied in the field (14, 16). Once rhizobia have successfully entered the plant root, elucidating the metabolism of undifferentiated rhizobia inside the nodule is important to avoid delays in the onset of nitrogen fixation.

Due to the complexity of nutrient exchanges in symbioses, metabolic modeling has become a popular tool for investigating rhizobium-legume interactions (3, 17). Metabolic models describe the reactions that are catalyzed by the enzymes annotated in an organism's genome (18, 19). By defining nutrient availability as well as an objective function reflecting the metabolic strategy of the organism, flux distributions at steady state can be calculated using flux balance analysis (20). Due to the gene-protein-reaction associations contained in metabolic models, they also provide a convenient framework for contextualizing genome-scale data obtained by omics technologies, such as transcriptomics or proteomics (21). Most metabolic models of rhizobial species so far have focused on fully differentiated bacteroids (22–26). One *in silico* study of *Sinorhizobium meliloti* has addressed the differences in metabolism for free-living growth in the bulk soil, growth of the rhizosphere, and symbiotic nitrogen fixation during the bacteroid stage (9). However, this study focused on the contributions of the different replicons to fitness in the different environments rather than specifics of changes in metabolic flux distributions and did not integrate experimental data. Another study compared the metabolism of free-living *Bradyrhizobium japonicum* with bacteroids (27). While transcriptome and proteome data sets were used to generate condition-specific models, the data for the free-living model were obtained for bacteria grown in a laboratory culture rather than the rhizosphere. Only one recent study has addressed metabolic differences in the different nodule zones for the symbiosis between *S. meliloti* and *Medicago truncatula* (28).

In this study, we reconstruct and extensively curate a genome-scale metabolic model (GSM) for *R. leguminosarum* bv. *viciae* 3841 (Rlv3841). Various experimental data sets exist for this strain at different stages of symbiosis with its native host pea. By integrating transcriptome, proteome, and gene essentiality data with the GSM, we perform a detailed investigation of nutrient uptake and metabolic pathway usage of Rlv3841 in the rhizosphere, as nodule bacteria and as nitrogen-fixing bacteroids. This genome-scale approach for data integration reproduced experimentally observed phenotypes and particularly highlighted the role of different carbon sources and amino acids throughout the different stages of symbiosis. The metabolic model developed herein provides a valuable resource for targeted investigation of metabolic requirements of

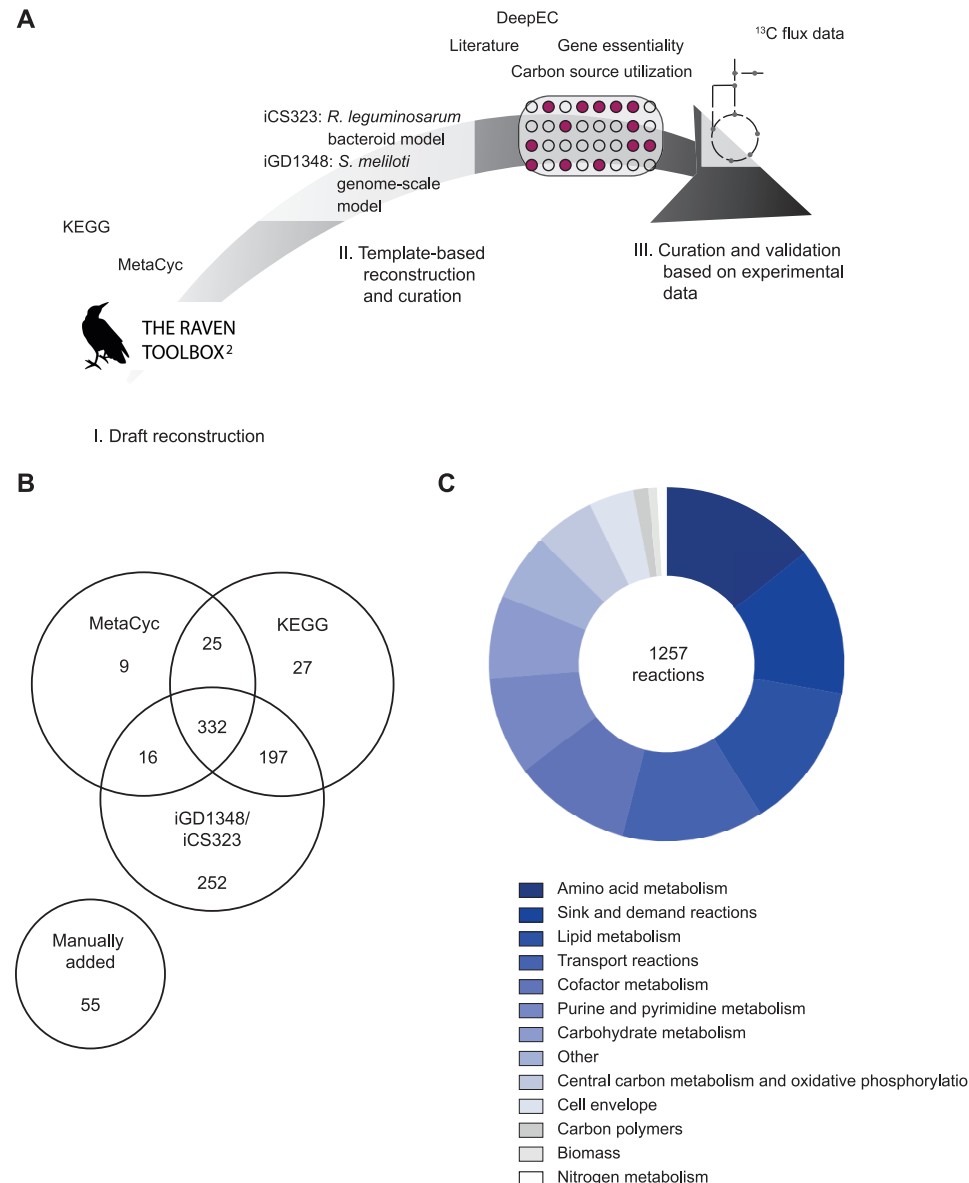

**FIG 1** Reconstruction of a genome-scale model for *Rhizobium leguminosarum* bv. *viciae* 3841. (A) Reconstruction process for iCS1224 using automated reconstruction, template-based reconstruction, and data-based curation. (B) Sources for the 913 metabolic reactions in iCS1224. Numbers indicate how many reactions from KEGG, MetaCyc, or the template-based reconstruction were included in the final model, with numbers in the overlapping areas indicating reactions that were present in multiple draft reconstructions. (C) Classification of the reactions in iCS1224.

the different rhizobial lifestyles and may enable the identification of strategies for engineering strains that are metabolically advantaged at all stages of symbiosis formation.

## RESULTS

**Reconstruction of a genome-scale metabolic model for *Rhizobium leguminosarum*.** Most published metabolic models for rhizobia focus on bacteroids and are therefore limited to metabolic pathways active during nitrogen fixation. Curated genome-scale reconstructions are so far only available for *B. japonicum* (27) and *S. meliloti* (28). With the aim of investigating metabolism in the rhizosphere and during different stages of bacteroid development, we developed a GSM for Rlv3841 using multiple sources of information. As shown in Fig. 1A, automated reconstructions based on the KEGG (29) and MetaCyc (30) databases were combined with a homology-based reconstruction

**TABLE 1** Properties of iCS1224

| Feature | Value |
|---|---|
| Genes | 1,224 |
| Metabolites | 984 |
| Unique EC identifiers | 603 |
| Reactions | 1,257 |
| Metabolic reactions | 913 |
| Gene-associated metabolic reactions | 897 |
| Transport reactions | 162 |
| Gene-associated transport reactions | 142 |
| Sink reactions | 155 |
| Demand reactions | 15 |
| Other reactions (e.g. DNA synthesis, protein synthesis, biomass objective function) | 12 |

using a GSM for *S. meliloti* as a template and reactions from our previously reconstructed bacteroid model of Rlv3841 (23) (Fig. 1B). Extensive curation was then performed based on literature evidence, gene essentiality data (13, 31) and enzymatic functions predicted by DeepEC (32). Comparison with iML1515, a high-quality model for *Escherichia coli* (33) as well as the CarveMe template for Gram-negative bacteria (34), was further used to correct reaction stoichiometry and reversibility if required. We next defined a biomass function based on evidence from the literature (Table S1). Because our previous work showed the dependence of carbon polymer synthesis on environmental conditions (23), demand reactions for polymers such as glycogen, polyhydroxybutyrate (PHB), and exopolysaccharides were included in the model to allow for their flexible accumulation. The final model contained 1,224 genes, 1,257 reactions, and 984 metabolites (Table 1), and was named iCS1224 according to standard naming conventions. The largest groups of metabolic reactions were associated with amino acid and lipid metabolism (14.1% and 13.5% of model reactions, respectively), followed by cofactor metabolism (10.7%) and purine/pyrimidine metabolism (9.0%) (Fig. 1C). Cluster of orthologous genes (COG) (35) analysis of the model genes showed that all COG categories associated with metabolic reactions were represented in iCS1224 (Fig. S1). The quality of the reconstruction was evaluated using MEMOTE (36), where iCS1224 achieved an overall score of 89%.

**Model validation.** We validated our model for free-living Rlv3841 growing in minimal media using various experimental data sets. First, we experimentally assessed growth on 190 different carbon sources using phenotype microarrays (37) (Data set S1). For the 109 carbon sources that were present as metabolites in iCS1224, an overall predictive accuracy of 89.9% with 90.9% precision and 96.4% recall was achieved (Fig. 2A), which is similar to the performance of curated GSMs for well-investigated bacteria, such as *Pseudomonas aeruginosa* (38) or *E. coli* (39). In addition, we evaluated the quality of gene essentiality predictions by comparing *in silico* gene essentiality with the results of an INSeq gene essentiality screen of Rlv3841 performed in minimal media supplemented with succinate and ammonia (31). Because the classification of genes based on transposon mutagenesis screen is subject to some variability (40), the list of essential genes was further curated by comparison with INSeq data for growth on complex media (13). Predictions by iCS1224 for gene essentiality during growth in minimal media achieved an accuracy, precision and recall of 91.0%, 89.6%, and 87.8%, respectively (Fig. 2B), thus showing good agreement with the INSeq data and indicating high quality of the gene-protein-reaction associations as well as suitability of the biomass objective function.

Finally, we performed quantitative validation of our model by comparing the predicted and experimentally measured growth rates in minimal media with glucose or succinate as the sole carbon source. After constraining the carbon uptake flux to experimentally determined values (41), the predicted growth rates were 0.150 h$^{-1}$ and

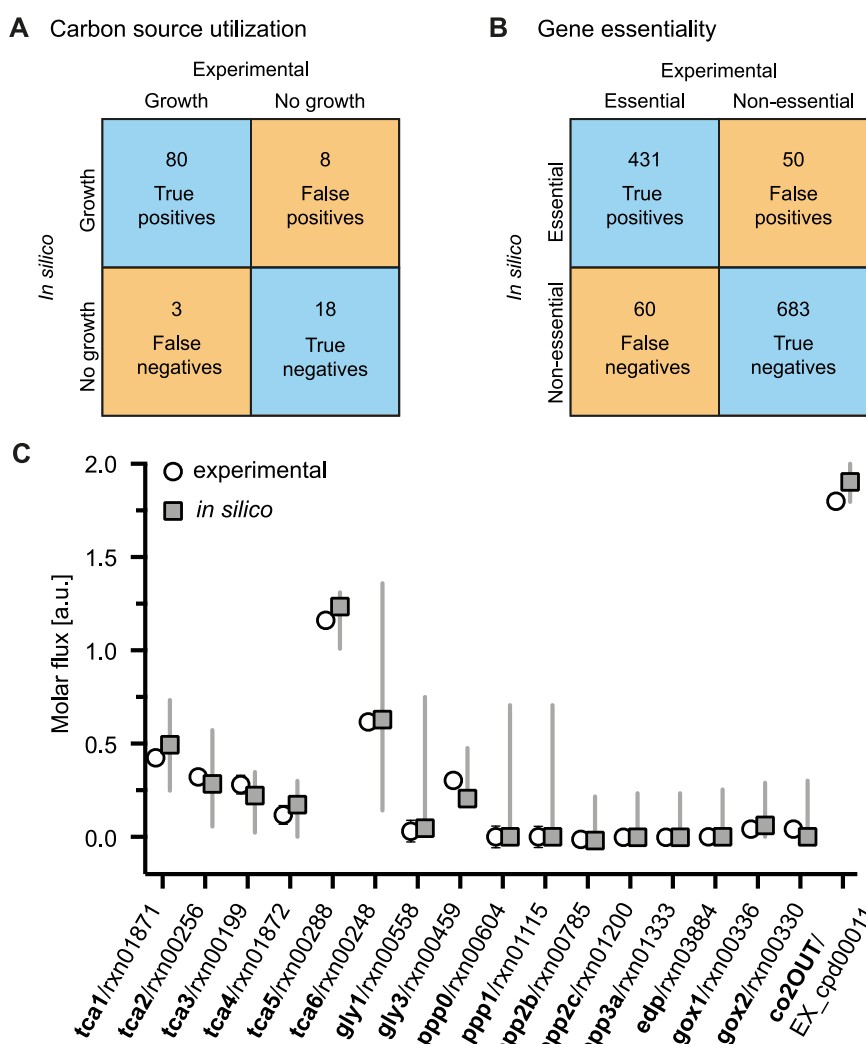

**FIG 2** Validation of iCS1224. (A) Table showing the agreement between carbon source utilization experimentally measured with phenotype microarrays and predicted by iCS1224. (B) Table showing the agreement between gene essentiality determined by insertion sequencing (13, 31) and predicted by iCS1224. (C) Comparison of metabolic fluxes determined by $^{13}$C metabolic flux analysis for Rlv3841 grown on succinate (43) with flux rates predicted by iCS1224. For experimental data, symbols and bars indicate mean ± SD. Note the error bars are too small to be visible for most data points. For *in silico* data, symbols represent the flux rate predicted by flux balance analysis, with lines indicating upper and lower bounds for each flux determined by flux variability analysis with at least 95% of the optimum flux through the biomass objective function. Labels on the *x* axis indicate the name of the reaction as reported in (43) (in bold), as well as the reaction identifier in the model.

$0.149 \text{ h}^{-1}$ for glucose and succinate, respectively. These values are consistent with previously reported growth rates, which range between $0.131 \text{ h}^{-1}$ and $0.187 \text{ h}^{-1}$ for glucose and between $0.102 \text{ h}^{-1}$ and $0.173 \text{ h}^{-1}$ for succinate (31, 41). While substantially slower growth of Rlv3841 on succinate compared to glucose was observed in one study (41), other studies reported similar growth rates for both carbon sources (31, 42), which agrees with our model predictions. Predicted flux values for 17 reactions involved in central carbon metabolism were further compared with published values measured by $^{13}$C metabolic flux analysis of Rlv3841 grown in minimal media with succinate and ammonia (43). As shown in Fig. 2C, we observed excellent agreement between predicted and experimentally measured flux values. In all cases, the measured flux was within the range determined by flux variability analysis. iCS1224 thus appears to be an accurate representation of the metabolism of Rlv3841, both qualitatively and quantitatively.

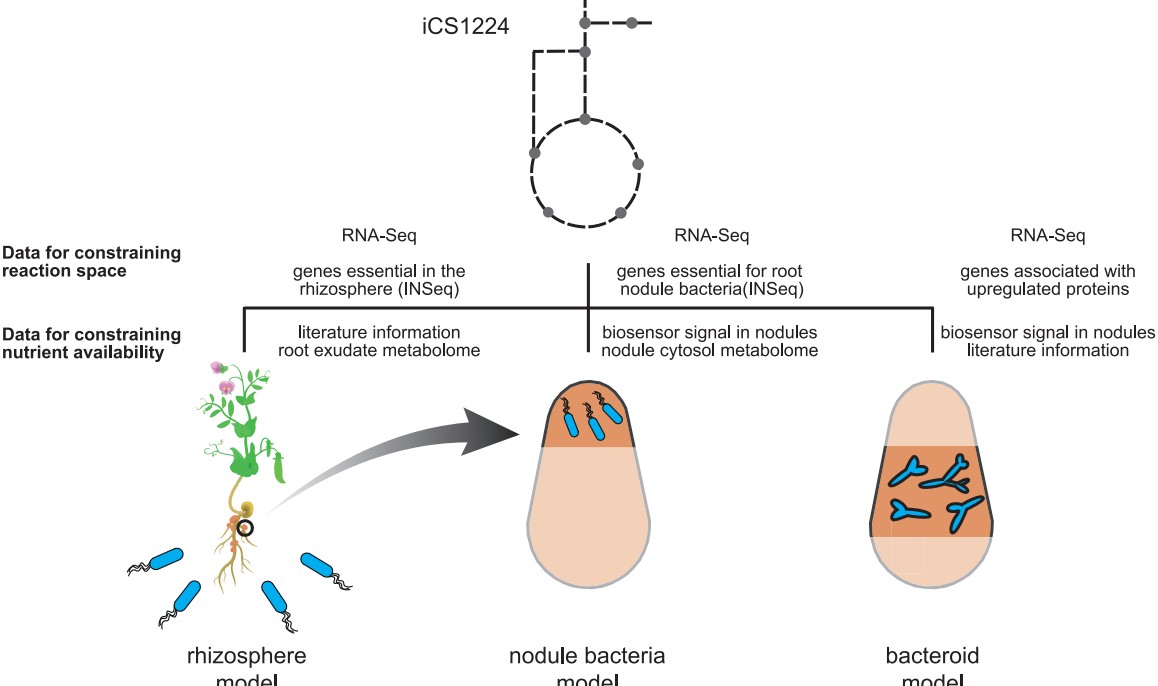

**FIG 3** Approach for generating lifestyle-specific models for *Rhizobium leguminosarum* bv. *viciae*. Based on iCS1224, transcriptome, gene essentiality, and proteome data specific to a certain lifestyle were used to inform the extraction of context-specific models for the rhizosphere, nodule bacteria, and nitrogen-fixing bacteroids. Boundary conditions were defined based on metabolome data and/ or literature information.

**Metabolism of rhizosphere bacteria.** Having validated the predictive capabilities of iCS1224, we sought to extract condition-specific models for metabolism of Rlv3841 (i) in the rhizosphere, (ii) as undifferentiated nodule bacteria, and (iii) as nitrogen-fixing bacteroids (Fig. 3). We chose the recently developed RIPTiDe algorithm (44) to obtain condition-specific metabolic models. Based on gene expression data, RIPTiDe assigns weights to all gene-associated reactions, assuming that higher transcript abundance makes it more likely that the corresponding reaction is used in a certain environmental condition. The overall flux through the network is then minimized and inactive reactions are removed. Finally, flux sampling of the solution space is performed, where flux through reactions associated with highly expressed genes is favored. In contrast to other methods for transcriptome data integration, RIPTiDe does not impose arbitrary thresholds on the gene expression data, it produces functional models with flux through the objective reaction, and takes flux parsimony into account, i.e., the overall flux is minimized to find cost-efficient solutions (44).

Generation of a rhizosphere-specific model thus required information about available nutrients as well as gene expression data. Nutrient availability in the rhizosphere is mainly determined by plant root exudates, and plants modulate the composition of their root exudates to select for specific soil microbes (45, 46). However, only a subset of metabolites is used by the soil microbiota (47, 48), and elucidation of nutrient uptake by rhizosphere bacteria usually requires extensive metabolomics profiling (49, 50). Taking a top-down approach for defining the rhizosphere environment, we first compiled a list of compounds present in pea root exudates based on published experimental data (10, 12, 51, 52) (Table S2). For those compounds that could be matched to model metabolites, exchange reactions were added to the model with reaction bounds set to allow for unlimited uptake. RNA-Seq data for Rlv3841 in the rhizosphere of pea plants 7 days postinoculation was used as an input data set for model contextualization. In addition, a list of genes that were classified as essential or defective in the rhizosphere in an INSeq screen (13) was provided to prevent removal of reactions

associated with these genes from the rhizosphere-specific model. Biomass production was set as the objective and additional positive lower bounds were placed on reactions involved in exopolysaccharide, lipopolysaccharide and Nod factor synthesis. All of these polymers are known to be important in the rhizosphere (4), and their forced production ensured inclusion of the respective biosynthetic pathways in the contextualized model. During data integration, constraining flux through the objective function to values between 50% and 95% of its maximum was tested to identify the scenario that gave the best match with the transcriptome data. Within the range of objective values tested, the highest correlation (Spearman's Rho = 0.237, $P < 0.001$) between metabolic fluxes and transcript abundances was obtained with the biomass reaction constrained to carry at least 77.5% of its maximum flux, and the rhizosphere-specific model contained 606 reactions and 576 metabolites. Remarkably, out of the 134 nutrients available for uptake before data integration, only 51 were present in the rhizosphere-specific model.

For the analysis of the contextualized model, we focused on those metabolic pathways that are either not universally essential or that are retained in the model despite their end product being available for uptake from the environment. Pathways such as membrane lipid or PHB synthesis, for instance, will always be retained in the model, because they are required to maintain flux through the biomass objective function and uptake of lipids and PHB is not possible. In addition, we limit our discussion to reactions that had a non-zero median flux value based on the flux sampling results, because those reactions are most likely to be active in the rhizosphere. The TCA cycle was predicted to be a central catabolic pathway (Fig. 4), which is consistent with previous reports of organic acids being the predominant carbon sources for rhizobia in the rhizosphere (9, 10). In particular, the model predicted high uptake of glycolate in agreement with the induction of C2 metabolism observed in previous gene expression studies (10). Glycolate was converted into pyruvate via glycolate oxidase and an aminotransferase. The model also showed high uptake rates for aspartate, which could explain the induction of a *dctA* biosensor in Rlv3841 in the pea rhizosphere (12). Aspartate and 2-oxoglutarate were transaminated to produce glutamate and oxaloacetate, which is a TCA cycle intermediate.

In addition to organic acids, amylotriose, which is hydrolyzed into glucose, was partly metabolized via the Entner-Doudoroff pathway and glycolysis in the model and entered the pentose phosphate pathway to enable production of nucleotides required for the synthesis of various polysaccharides and Nod factors. The gene encoding the solute-binding protein of a carbohydrate uptake transporter-1 (CUT1) family transporter (RL3840) was 2.6-fold upregulated in the pea rhizosphere compared to free-living cells (10), which supports the predicted uptake of a di- or oligosaccharide. Ribulose, a monosaccharide metabolized via the pentose phosphate pathway was also predicted to be taken up. Catabolism of a monosaccharide in the rhizosphere is highly probable considering the strong signals of a fructose and a xylose biosensor in the pea rhizosphere (12). The fructose biosensor is based on the solute-binding protein of the CUT2 family *frcABC* transporter, which has been shown to transport ribose in addition to fructose in *S. meliloti* (53) and may therefore also contribute to pentose uptake in the rhizosphere. The model further contained reactions for glycerol uptake and catabolism, which could explain the decreased competitiveness observed for a glycerol catabolism mutant of *R. leguminosarum* bv. *viciae* VF39 (54).

With regard to amino acids, all of which are present in root exudates, biosynthetic pathways were generally retained in the rhizosphere model due to the essentiality of the associated genes. Low levels of uptake were however predicted for most amino acids, mainly to support protein synthesis. Notably, the biosynthetic pathway for leucine was predicted to be active, which was partly supported by uptake of 2-isopropylmalate, an intermediate of branched-chain amino acid synthesis. The need for leucine synthesis in the rhizosphere agrees with a *leuD* mutant of Rlv3841 requiring the addition of 1 mM leucine to nodulate pea (55). Mutation of the isopropylmalate synthase

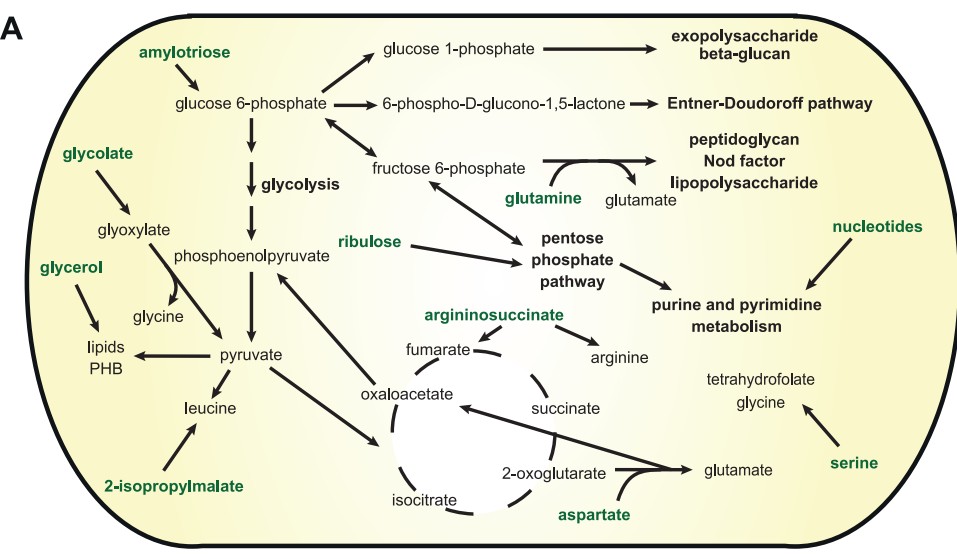

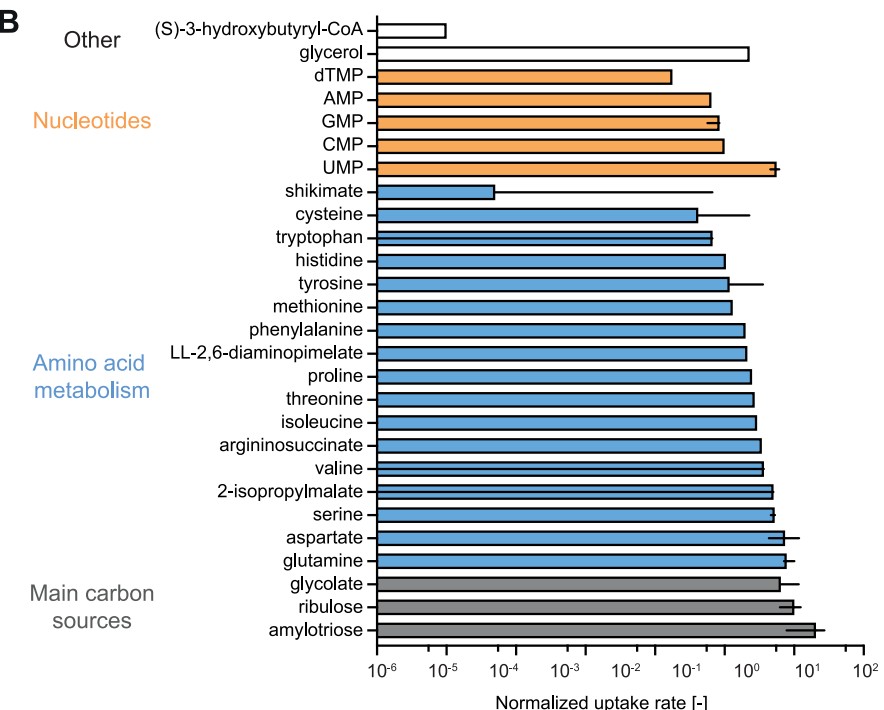

**FIG 4** Metabolism of *Rhizobium leguminosarum* in the pea rhizosphere. A rhizosphere-specific model was extracted from iCS1224 using the RIPTiDe algorithm with RNA-Seq and gene essentiality data for *R. leguminosarum* in the rhizosphere of pea plants. (A) Schematic representation of the main pathways predicted to be active in the rhizosphere-specific model. Compounds predicted to be taken up are indicated in bold green. Note that the magnitude of flux is not indicated in this summary map. (B) Bar graph showing the uptake rates of metabolites predicted to be taken up from pea root exudate. Absolute flux values for the exchange reactions were normalized by flux through the biomass reaction in each sample. Only metabolites with non-zero median uptake for the 500 samples of the contextualized model are shown. Uptake of ions and cofactors has been omitted for clarity. Bars and lines indicate median and interquartile range, respectively.

gene in *S. meliloti* impaired nodulation even in the presence of leucine, and it was shown that either isopropylmalate synthase or intermediates of the leucine biosynthetic pathway are required for the activation of *nod* gene expression (56). It is therefore possible that the predicted leucine synthesis is at least partly related to the synthesis of Nod factors in the rhizosphere. High uptake rates were further predicted for glutamine, which is consistent with its high concentration in pea root exudates (52).

Glutamine was converted into glutamate, which was mostly used to sustain leucine synthesis. The model also contained active uptake reactions for several nucleotides. This agrees with the reported uptake of nucleosides and nucleotides by rhizosphere bacteria (47, 50, 57) and agrees with the gene essentiality predictions for rhizosphere bacteria obtained by INSeq, where purine auxotrophs appear to be rescued by plant root exudates (13).

Among biomass components that are present in root exudate but not predicted to be taken up, biosynthesis of the polyamine putrescine was retained in the model, attesting to the ability of the RIPTiDe algorithm to choose metabolic reactions that agree with gene expression and/or essentiality rather than choosing the least resource-intensive solution. Putrescine and related polyamines are important for survival under stress conditions and their synthesis has been suggested to play an important role during root colonization (58). As part of the model reconstruction, several demand reactions were included for compounds such as carbon polymers whose accumulation can vary with environmental conditions. The only non-essential demand reactions that were not removed during the pruning process were those for glutathione and polyphosphate, where polyphosphate synthesis in particular had a non-zero median flux. Glutathione is important to deal with stress conditions, such as osmotic and oxidative stress, encountered in the rhizosphere, and mutants in glutathione biosynthesis are severely affected in rhizosphere colonization (41). The predicted catabolism of glycolate via glyoxylate produces the reactive oxygen species hydrogen peroxide, which could contribute to the need for glutathione synthesis. Polyphosphate has recently been suggested to play a role in the global carbon regulatory system (59), but its function remains to be investigated in detail. It is interesting to note that an exopolyphosphatase gene (RL1600) was classified as essential for persistence in the rhizosphere (13), indicating an important role for phosphate homeostasis in rhizosphere colonization and/or competition.

Catabolism of several other compounds, such as erythritol, *myo*-inositol and homoserine, has been described to be important for competitiveness (10, 60, 61); however, these compounds were not included in the rhizosphere model. This could be due to the catabolism of these compounds being important at later stages of the symbiosis, e.g., for growth in infection threads rather than in the rhizosphere. Alternatively, uptake of these compounds could be masked in the model due to catabolic routes that are shared with other metabolites. For example, erythritol is metabolized via the pentose phosphate pathway (62); hence the predicted uptake and metabolism of ribulose could partly be due to erythritol catabolism.

**Reporter metabolites highlight plant-specific rhizosphere metabolism.** As an independent validation and extension of our analysis of metabolic changes, we identified reporter metabolites using previously published microarray data comparing Rlv3841 in the rhizosphere of pea plants with free-living cells grown on minimal media with glucose and ammonium chloride (10). Based on the network topology defined by a metabolic model, the reporter metabolite algorithm identifies those compounds around which significant changes in gene expression occur (63). This method is therefore independent of specifying nutrient uptake from the environment. Reporter metabolites associated with upregulated genes matched several observations from the RNA-Seq data integration described in the previous section. In particular, several intermediates of branched-chain amino acid synthesis, such as acetolactate, 2-hydroxyethyl-thiamine diphosphate and 2-aceto-2-hydroxybutanoate, were identified as reporter metabolites (Fig. 5A). Significant transcriptional changes were also observed around various nucleobase derivatives. This may be related to their predicted uptake from plant root exudates but could also indicate an increased need for nucleotide synthesis for the production of polysaccharides and Nod factor. Phosphoribosyl-AMP and phosphoribosylformiminoaicar-phosphate are intermediates of histidine biosynthesis and direct precursors of AICAR, which is involved in purine metabolism. Because no additional metabolites of the histidine biosynthetic pathway were identified as reporter metabolites, this analysis indicates an increase in

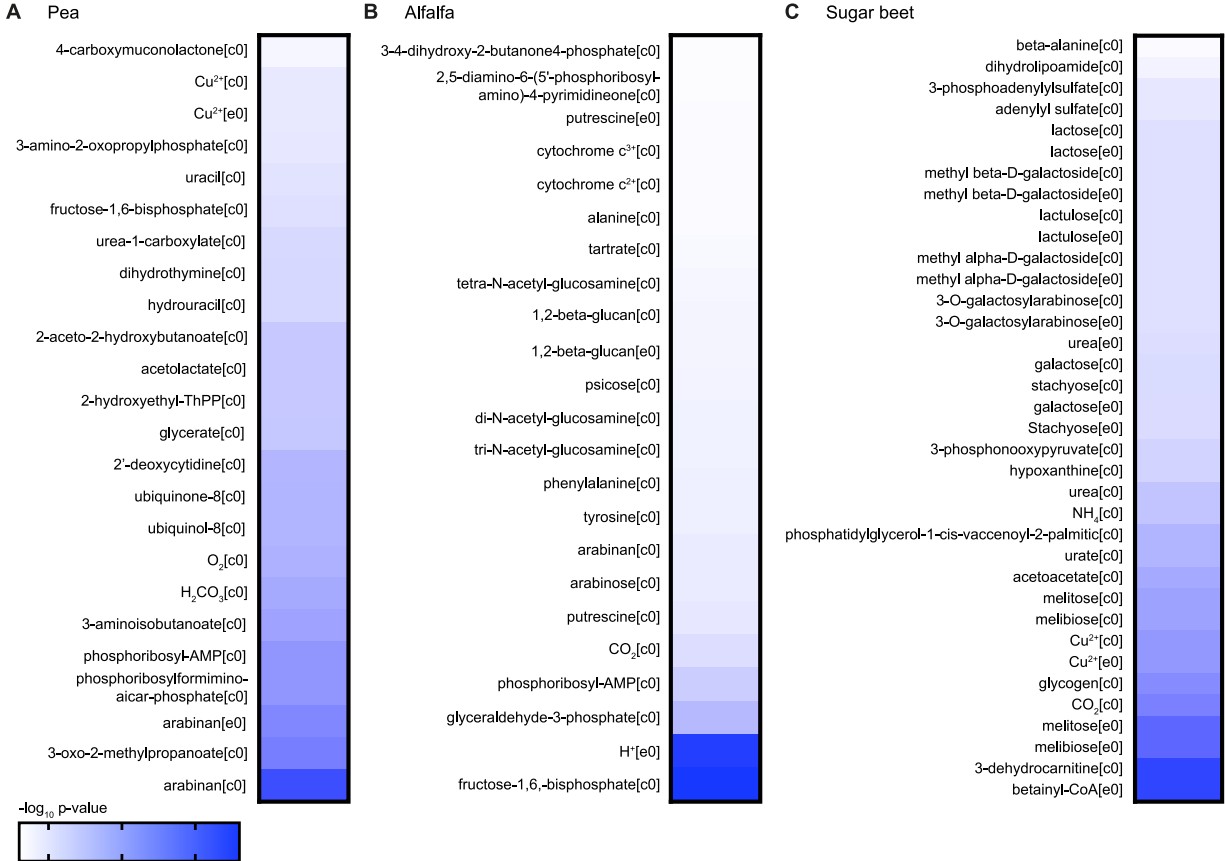

**FIG 5** Reporter metabolites in different rhizospheres. Reporter metabolites were calculated using microarray data for *Rhizobium leguminosarum* bv. *viciae* 3841 in the rhizosphere of pea (A), alfalfa (B), and sugar beet (C) compared with free-living cells grown in minimal media with glucose and ammonia (10). The heatmaps show the negative decimal logarithm of the $P$ value for those metabolites that were associated with significant ($P < 0.05$) transcriptional changes among genes upregulated in the rhizosphere. [c0] and [e0] indicate cytosolic and extracellular metabolites, respectively.

AICAR synthesis, which seems to be required for successful legume infection by various *Rhizobium* species (64).

Comparison of the rhizosphere reporter metabolites for pea (host legume for Rlv3841) with those for alfalfa (non-host legume) (Fig. 5B) and sugar beet (non-legume) (Fig. 5C) highlighted several plant-specific features. For alfalfa, phosphoribosyl-AMP was identified as a reporter metabolite similar to pea. In addition, phenylalanine and tyrosine support the role of aromatic amino acid metabolism in colonization competitiveness (10). Significant transcriptional changes also occurred around the carbon polymer beta-glucan and the diamine putrescine. While beta-glucan generally appears to be important for persistence in the rhizosphere (10, 13), its identification as a reporter metabolite together with putrescine indicates increased osmotic stress in the alfalfa rhizosphere compared to pea. For sugar beet, the identification of several compounds involved in nitrogen metabolism (ammonia, urea, urate) agrees with the suggested nitrogen limitation in the sugar beet rhizosphere, but nitrogen sufficiency in legume rhizospheres (10). This could also explain why the carbon polymer glycogen was a reporter metabolite specifically in the sugar beet rhizosphere because glycogen synthesis is probably linked to nitrogen limitation (65). Notably, multiple mono- and disaccharides and their derivatives indicate an increased importance of sugar metabolism compared with legume rhizospheres. However, many genes involved in saccharide metabolism are associated with multiple reactions (e.g., unspecific glucoside hydrolases), and therefore the identity of the metabolized sugar cannot be derived from this analysis. Finally, the reporter metabolites 3-dehydrocarnitine and betainyl-CoA indicate

either accumulation of amines for osmoprotection or catabolism of carnitine or related amines. These findings present interesting targets for future investigations using gene essentiality screens on different plant hosts.

Overall, both the context-specific model obtained by transcriptome data integration and the reporter metabolite analysis were in good agreement with experimental data for rhizobial metabolism in the rhizosphere without forcing the uptake of any compound through arbitrary constraints. Instead, insights into nutrient uptake were facilitated by the integration of gene expression and gene essentiality data with iCS1224. If biomass production were simply maximized with unlimited availability of all root exudate compounds, this would result in uptake of all available compounds that are required for biomass formation, which would not reflect a biologically meaningful scenario.

**Metabolism of undifferentiated nodule bacteria.** We next sought to develop models for Rlv3841 inside the nodule environment. For this purpose, it is important to differentiate between nodule bacteria at the tip of the nodule, which are dividing and undergoing differentiation, and bacteroids in the central nitrogen fixation zone of the nodule (66). While nodule bacteria are still dividing, bacteroids are growth-arrested and mainly catabolize plant-provided dicarboxylates to fix atmospheric $N_2$ into ammonia. The distinction between these developmental stages is required in the context of gene essentiality analyses because genes required for the differentiation process may not be essential for nitrogen fixation and vice versa. Similar to the approach for the rhizosphere model, we used RIPTiDe to obtain models for nodule bacteria and bacteroids and performed flux sampling to identify those reactions that are most likely to be active in each contextualized model.

The model for nodule bacteria was obtained using published dRNA-Seq data for RNA extracted from nodule tips (67), as well as a list of genes that were predicted to be specifically essential for nodule bacteria (13). Nutrient availability was defined based on a study using biosensors to detect metabolites inside nodules (12) and our direct measurement of metabolites in pea root exudate, in pea bacteroids and in the nodule cytosol as described previously (43) (Table S3, Data set S2). The biomass objective function was used to account for the cell division occurring as rhizobia grow down infection threads and differentiate into bacteroids and positive lower bounds were placed on demand reactions for exopolysaccharides and lipopolysaccharides. The nodule bacteria model contained 510 reactions and 502 metabolites and achieved highest correlation with the transcriptome data (Spearman's Rho = 0.335, $P < 0.001$) when the objective value was constrained to 65% of its maximum. The observation that higher correlation was obtained for lower flux through the objective reaction (compared to the rhizosphere) indicates that the metabolism of nodule bacteria is not oriented toward maximum growth. This agrees with experimental data showing that growth of infection threads proceeds at highly variable rates controlled by the plant host (68). The improved correlation of flux predictions and gene expression data compared with the rhizosphere model can be explained by the lower number of essential genes, which places fewer constraints on the reactions included in the contextualized model.

Malate, fructose, xylose, *myo*-inositol, and $\gamma$-aminobutyrate (GABA) were all predicted to be taken up by nodule bacteria (Fig. 6A and Fig. S2). Biosensors for these carbon sources were strongly induced in young nodules, whereas biosensors for the carbon sources that were removed during the data integration process (erythritol, mannitol, formate, malonate, tartrate) only showed weak induction (12). Malate and GABA are both catabolized in the TCA cycle, indicating that it is an important catabolic route in differentiating nodule bacteria despite transport of dicarboxylates being nonessential for differentiation into bacteroids (69). Enzymes involved in GABA metabolism are highly induced in bacteroids, although GABA catabolism is not essential for effective nitrogen fixation (70). The predicted catabolism of fructose is consistent with the strong induction of a fructose-specific biosensor in nodules (12) as well as a previous modeling study of *S. meliloti* suggesting the use of sucrose-derived sugars as a carbon

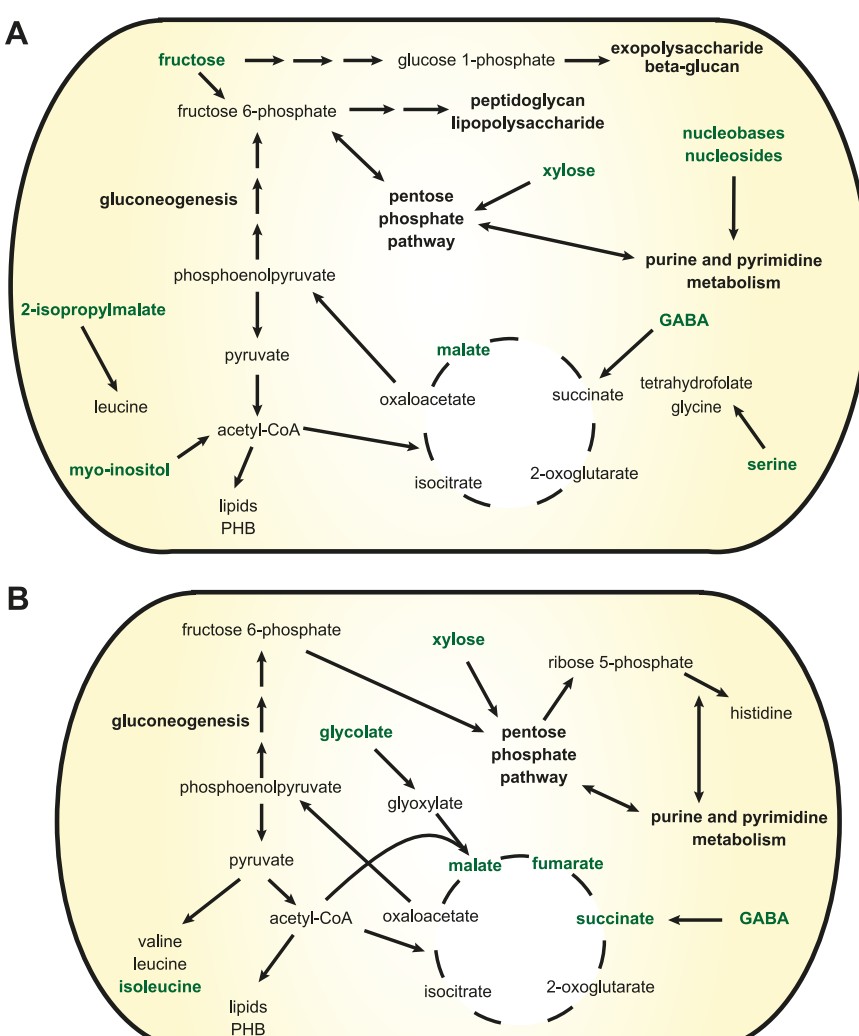

**FIG 6** Metabolism of undifferentiated nodule bacteria and nitrogen-fixing bacteroids. Maps showing schematic representations of the main pathways predicted to be active in undifferentiated nodule bacteria (A) and nitrogen-fixing bacteroids (B). Compounds predicted to be taken up are indicated in bold green. Note that the magnitude of flux is not indicated in these summary maps.

source by differentiating nodule bacteria (28). Sucrose uptake was removed from the nodule bacteria model, which may be due to the inability of the model to accurately distinguish between sucrose and fructose uptake based on the gene expression data. It is interesting to note that Rlv3841 bacteroids mutated in a subunit of succinyl-CoA synthetase, which had severely reduced nitrogen fixation capacity, had 168-fold higher levels of fructose than wild-type bacteroids and 151-fold elevated levels of sucrose (Data set S2), which may be the result of carbon source build-up in the developmentally impaired nodule bacteria and bacteroids. *Myo*-inositol is present in the rhizosphere (12) and abundant in pea nodules (71), and mutants in *myo*-inositol catabolism have strongly reduced competitiveness compared with wild-type Rlv3841 (60). However, the activity of enzymes involved in *myo*-inositol catabolism is very low in mature bacteroids (71), and mutants in *myo*-inositol catabolism were not disadvantaged during growth in the rhizosphere compared to wild-type Rlv3841 (60). In addition, it has been proposed that catabolism of rhizopines, which are inositol derivatives, by undifferentiated nodule bacteria may be important as a kin selection strategy (72). Catabolism of *myo*-inositol is therefore most likely to play a role during infection and in undifferentiated nodule bacteria, which is correctly predicted by the model. Xylose enters the pentose phosphate pathway, and its predicted uptake could be related to

the importance of nucleotide synthesis, both for DNA endoreduplication and synthesis of exopolysaccharides and lipopolysaccharides. Similarly, uptake reactions for the nucleoside guanosine and uridine as well as the nucleobase adenine were present in the model.

Our nodule bacteria model predicted uptake of most amino acids, which agrees with the severe symbiotic defect of a *gltB* mutant unable to transport amino acids (73) but may also be a result of a beginning general downregulation of biosynthetic functions as rhizobia transition into growth-arrested bacteroids. Similar to the rhizosphere bacteria, leucine was predicted to be synthesized from 2-isopropylmalate. Expression of *nod* genes is elevated in nodule bacteria at 7 days postinoculation (74), which could explain the predicted leucine synthesis as discussed for the rhizosphere model.

**Metabolism of bacteroids.** To extract a model specific for nitrogen-fixing bacteroids, we used dRNA-Seq data derived from the middle of nodules (67), which contains fully differentiated bacteroids performing nitrogen fixation (66). In addition, a list of 38 genes that were present in the model and encoded proteins significantly upregulated in bacteroids compared to free-living bacteria (23) and the *dct* genes (75) were specified to ensure inclusion of those genes in the bacteroid model. Nitrogenase activity was set as the objective function while low levels of protein and fatty acid production were enforced through demand reactions. Nutrient availability was specified similar to the considerations for nodule bacteria (Table S4). Gene essentiality data from the INSeq screen were not included for model contextualization due to the aforementioned difficulty of determining the developmental stage where a gene is essential inside the nodule environment. In contrast to our previously reconstructed model for bacteroid metabolism (iCS323) (23), the bacteroid model was thus obtained using a top-down approach to constrain iCS1224 rather than assembling individual pathways in a bottom-up manner, and uptake of a wider range of nutrients was enabled. The bacteroid model contained 307 reactions and 308 metabolites and achieved significant correlation with the transcriptome data (Spearman's Rho = 0.348, $P < 0.001$) when nitrogenase activity was constrained to 65% of its maximum. The reduced model size compared to both the rhizosphere and the nodule bacteria model is in agreement with the reduced physiological complexity of the non-dividing bacteroids (3, 74). Out of the 236 metabolic reactions in the bacteroid model, 137 (58.1%) are not present in iCS323. This difference is partly due to a more detailed representation of fatty acid biosynthesis in the bacteroid model derived from iCS1224, accounting for 50 reactions that are not shared with iCS323. In addition, pathways for the synthesis of cofactors such as cobalamin, ubiquinone, and heme comprise 53 reactions in the bacteroid model derived from iCS1224 but are not included in iCS323. Out of the 108 metabolic reactions that are present in iCS323 but absent from the bacteroid model derived from iCS1224, 64 are involved in amino acid biosynthesis because amino acid uptake was not permitted in iCS323.

The bacteroid model contained the C4 dicarboxylates malate, succinate, and fumarate as the main carbon sources in agreement with experimental evidence (42, 69) (Fig. 6B and Fig. S3), and only low levels of GABA uptake were predicted. Ammonia was the only nitrogenous export product. Consistent with our previous modeling study of Rlv3841 bacteroids (23), constraining the oxygen uptake prior to data integration resulted in nitrogen partly being secreted as alanine. With the metabolites provided in initial simulations, the glyoxylate cycle comprising isocitrate lyase and malate synthase was contained in the model, which is consistent with the high induction of malate synthase (74) but disagrees with the lack of isocitrate lyase activity in pea bacteroids (76). The source of glyoxylate for the malate synthase reaction has so far not been elucidated. Because the metabolomics data showed that glycolate is present in the nodule cytosol and glycolate concentrations in bacteroids are 2-fold elevated compared to free-living cells (Data set S2), we allowed for glycolate uptake by the bacteroid model and inactivated the isocitrate lyase reaction. This resulted in substantial uptake of glycolate, which was converted into glyoxylate that was used in the malate synthase

reaction. Glycolate provision by the plant may therefore explain the increase in malate synthase expression in the absence of isocitrate lyase activity.

The model also predicted uptake of xylose, which was metabolized in the pentose phosphate pathway to support synthesis of nucleotides. Dicarboxylate catabolism generally requires gluconeogenesis to provide precursors for the synthesis of nucleotides and some amino acids. Due to the predicted uptake of xylose, only minor fluxes through the reactions involved in gluconeogenesis occurred, highlighting the importance of this pathway in bacteroids as an interesting question to explore using targeted mutant studies. Proton uptake by bacteroids was required as previously predicted for *S. meliloti* bacteroids (28) and a demand reaction for PHB was retained in the model. PHB synthesis was highly variable across flux samples, which is in agreement with its previously suggested role for carbon and redox balancing (23, 43).

Low levels of uptake were predicted for most amino acids to support the required synthesis of protein, but no significant catabolism of any amino acid was observed. Mutant studies have shown a requirement for branched-chain amino acid supply to bacteroids (55), and the model predicted that isoleucine is supplied by the plant. Interestingly, histidine was predicted to be synthesized rather than taken up by bacteroids. Several proteins involved in histidine synthesis were upregulated or unchanged in abundance in the bacteroid proteome compared to free-living Rlv3841 (23), in contrast to the general downregulation of amino acid biosynthesis (3). Similar results were obtained in a proteome study of *Rhizobium etli* (24) and RNA-Seq data for bacteroids of *R. leguminosarum* bv. *viciae* A34 and *R. leguminosarum* bv. *phaseoli* 4292 (77). In addition, mutants of *R. leguminosarum* bv. *trifolii* lacking histidinol dehydrogenase activity formed ineffective nodules on clover (78). To investigate the requirement for histidine biosynthesis, we compared the amino acid composition of the Nif and Fix proteins, which are highly expressed in bacteroids, with the overall amino acid composition of the Rlv3841 proteome (Table S5). We found a significant ($P = 0.042$) enrichment of histidine in the Nif and Fix proteins, which could at least partly explain why histidine biosynthesis is required in bacteroids.

## DISCUSSION

In this study, we present the first curated GSM for Rlv3841, a model strain for investigating rhizobium-legume interactions and a natural symbiont of the agriculturally important crop pea. GSMs have emerged as promising tools for informing experimental design, addressing fundamental research questions, and contextualizing experimental data (79). In order to obtain a high-quality model, integration of experimental data during model curation and validation is essential (80). We therefore evaluated our model using carbon source utilization, gene essentiality data and flux data obtained by $^{13}$C labeling and observed high agreement between model predictions and experimental data.

We further used the GSM to elucidate metabolic changes in Rlv3841 as it transitions from a free-living soil bacterium in the rhizosphere to an undifferentiated nodule bacterium and finally to a nitrogen-fixing bacteroid. While significant advances in determining metabolic requirements for successful symbiosis formation have been made using transcriptome data (10) and gene essentiality screens (13), genome-scale data sets are often difficult to interpret without the framework of a comprehensive model, especially when information about nutrient uptake is missing. To this end, we employed approaches integrating gene expression and metabolome data as well as gene essentiality predicted by INSeq to obtain condition-specific models. This allowed us to contextualize our model based on experimental data without assuming uptake rates for any nutrient. In addition, during the process of data integration, different fractions of the optimum objective value were tested as constraints to find a solution with the highest correlation between gene expression and associated reaction fluxes. Especially for nodule bacteria and bacteroids, using sub-optimal fluxes

through the objective function as constraints during model contextualization was found to produce better agreement with experimental data. Objective functions can be difficult to define outside of defined growth in a laboratory culture, and our results highlight the need to adopt strategies beyond maximization of a biomass objective function to accurately capture metabolic behavior in complex environmental settings. A clear limitation of our approach is the imperfect correlation of gene expression and protein abundance, as well as protein abundance and enzyme activity (81, 82). Catabolic pathways common to multiple different compounds can further make it difficult to specifically determine which nutrient is taken up. Nevertheless, our model predictions are in good agreement with known metabolic characteristics of the different lifestyles of Rlv3841, attesting to the biological relevance of our findings.

The rhizosphere model showed substantial uptake of glycolate, aspartate, and glutamine as well as mono- and oligosaccharides. These predictions are consistent with our previous transcriptional study of Rlv3841 in the rhizosphere of pea, alfalfa, and sugar beet (10) as well as nutrient uptake of a *Rhizobium* sp. from root exudates of *Arabidopsis* (47). Similarly, a model of *S. meliloti* predicted the importance of gluconeogenesis in the rhizosphere due to uptake of organic acids (9). We further identified a requirement for leucine synthesis in the rhizosphere, as well as a potentially important role for polyphosphate synthesis. However, the predicted nutrient uptake was not supported by gene essentiality predictions in all cases. While both genes predicted to be essential by INSeq and our metabolic model generate predictions that warrant detailed investigation using isolated mutant strains, there are other possible explanations for this observation. First, root exudates might contain insufficient quantities of a compound to complement an auxotrophy. In addition, the composition of plant root exudates changes over time (50), and compounds present at the time of RNA extraction may not be present at inoculation, causing the loss of some mutants. Finally, for genes that are essential on complex media, the corresponding mutants may already be lost from or underrepresented in the bacterial population inoculated onto plants.

The model for nodule bacteria confirmed previous results suggesting supply of nutrients other than dicarboxylates, in particular sucrose-derived sugars, to *S. meliloti* during the differentiation process (28). Interestingly, we found that *myo*-inositol catabolism was only predicted for nodule bacteria, but not in the rhizosphere or in bacteroids. While the importance of *myo*-inositol catabolism for competitiveness has been established (60), our results suggest that it may be particularly important for differentiating bacteria rather than those in the rhizosphere. In contradiction to our model for Rlv3841, *myo*-inositol catabolism in bacteroids was predicted for *R. etli* (26), *S. meliloti* (9), and *S. fredii* (22). However, given the absence of transcriptional upregulation of *myo*-inositol catabolic genes (74) and enzyme activity in Rlv3841 bacteroids (71), our model is consistent with experimental data, suggesting that *myo*-inositol catabolism in bacteroids may only occur in some symbioses. For bacteroids, biosynthesis of histidine was found to be important in contrast to the general uptake predicted for most other amino acids. In addition, low levels of xylose uptake were predicted to support nucleotide synthesis in bacteroids. This result indicates that a carbon source metabolized in the pentose phosphate pathway may be provided to bacteroids, which presents an interesting area to explore experimentally using mutants affected in gluconeogenesis. Substantial activity of the pentose phosphate pathway in bacteroids has also been predicted for *S. meliloti* (9) and *R. etli* (24), but only small fluxes through this pathway were predicted for *S. fredii* (22), indicating strain- and/or host plant-specific differences. Initial predictions of isocitrate lyase activity, which disagree with measured enzyme activities in bacteroids, led us to hypothesize that glycolate is provided to bacteroids. This is supported by metabolomics data and could explain the induction of malate synthase in bacteroids without concomitant expression of isocitrate lyase.

In summary, our results provide insights into rhizobial metabolism in the rhizosphere, which can inform the design of more competitive rhizobial inocula as well as plants that secrete metabolites to specifically enrich beneficial bacterial strains. Our understanding of the nutrient exchanges between plants and rhizobia at different developmental stages inside nodules remains incomplete (3, 83), and the predictions presented herein provide a foundation for targeted investigation of amino acid and central carbon metabolism in particular. We anticipate that the highly curated metabolic model for Rlv3841 presented in this article will provide a valuable resource for the reconstruction of GSMs for related species.

## MATERIALS AND METHODS

**Model reconstruction.** To reconstruct a GSM for Rlv3841, we combined information from multiple databases, which has been shown to significantly improve the scope of metabolic network reconstructions (84). All reconstructions were performed based on RefSeq assembly GCF_000009265.1. We used the RAVEN Toolbox 2.0 (85) to create draft models from KEGG (29) and MetaCyc (30) using the functions `getKEGGModelForOrganism` and `getMetaCycModelForOrganism`, respectively. In addition, template-based reconstruction based on BLAST bidirectional hits was performed using a curated GSM for *S. meliloti* 1021 (iGD1348 [28]) as a template for the function `getModelFromHomology`. All models were merged into one reaction list and reaction and compound identifiers were unified based on the reaction database provided by the ModelSEED (86), followed by removal of duplicate reactions. Starting from this database of reactions compiled from different sources, the reconstruction was curated. First, reactions without gene association were removed. Reactions involving nonspecific compounds such as "acceptor" or "protein" were also deleted, as well as reactions involved in the biosynthesis and catabolism of secondary metabolites and non-metabolic processes, such as DNA and RNA modification because those were outside of the scope of our model. Extensive curation was then performed by evaluating metabolic pathways guided by the literature and the KEGG database. Pathways for catabolism of small carbon sources in particular were reconstructed based on predictions obtained from GapMind (87). Gene-protein-reaction associations were curated based on published gene essentiality data for growth in minimal (31) and complete (13) media as well as enzyme commission (EC) number predictions obtained from DeepEC (32).

Transport reactions were annotated based on literature evidence, in particular homology to experimentally characterized transporters in *S. meliloti* (88) and the annotation obtained from TransportDB 2.0 (89). We manually reconstructed pathways for organism-specific biomass components, such as lipopolysaccharides and exopolysaccharides, as well as pathways which were not present in any of the databases used for reconstruction, such as carnitine metabolism. To improve information on reaction directionality, upper and lower bounds were adjusted according to the information in a highly curated model for *E. coli* (iML1515 [33]) and the CarveMe template model for Gram-negative bacteria (34).

The biomass objective function was defined as follows: The composition of DNA was determined from the RefSeq genome sequence. Similarly, RNA and protein composition were determined by counting nucleotides or amino acids in the annotated RNAs and protein coding sequences, respectively. Because the lipid composition of Rlv3841 has not been investigated so far, we adopted the values reported *R. leguminosarum* bv. *trifolii* ANU843 (90). *R. leguminosarum* produces predominantly C18 fatty acids, as well as smaller quantities of C16 fatty acids (43, 91), and representative phospholipids in our model included fatty acids with these chain lengths. Lipopolysaccharides and exopolysaccharides were included with the fractions previously reported for *S. meliloti* (92). Cyclic beta-glucans have so far not been considered in metabolic models for rhizobia; however, they can make up a significant fraction of the cellular dry weight (93) and were therefore also included as a biomass component. Apart from the main cell components, trace amounts of cofactors identified as universally essential in prokaryotes (94) were included in the biomass objective function. Phytoene was also added to the biomass reaction due to the essentiality of the genes associated with its biosynthetic pathway. Carbon polymers such as glycogen, PHB, and fatty acids, as well as polyamines such as homospermidine and putrescine, are known to be produced by Rlv3841; however, the quantities in which they are produced vary depending on nutrient availability. Similar to our previous work (23), we therefore added demand reactions for these compounds to allow for variable accumulation. Glycogen and PHB were also included in the biomass objective function as they are commonly synthesized by free-living *R. leguminosarum* (95). A complete description of the biomass composition used in this study is given in Table S1.

**General modeling procedures.** Standard metabolic modeling computations were performed in MATLAB R2020b (Mathworks) using scripts from the COBRA Toolbox v3.0 (96) and the Gurobi 9.1.1 solver (www.gurobi.com). When using the `optimizeCbModel` function, the Taxicab norm was minimized to avoid loops in the calculated flux distributions. All scripts are available on Github (https://github.com/CarolinSchulte/Rlv3841-lifestyles).

**Model validation.** To evaluate the agreement between model predictions and experimentally determined carbon source utilization, we limited our analysis to those compounds that were either present in the model or showed a positive growth phenotype in the phenotype microarray experiment. The lower bounds for the exchange reactions were then adjusted according to the composition of universal minimal salts (UMS) media (31) with ammonium as a nitrogen source, and flux through the biomass reaction

was evaluated for each carbon source individually added to the model. Accuracy, precision, and recall for carbon source utilization and gene essentiality analysis were calculated according to the following equations:

$$accuracy = \frac{TP + TN}{TP + TN + FP + FN}$$

$$precision = \frac{TP}{TP + FP}$$

$$recall = \frac{TP}{FP + FN}$$

TP: true positives    FP: false positives
TN: true negatives    FN: false negatives

Gene essentiality analysis was performed using the function `singleGeneDeletion` with the minimization of metabolic adjustment (MOMA) option in the COBRA Toolbox, while all components of UMS media with succinate and ammonia were available without constraints on their uptake rate. The predictions were compared with gene essentiality data for Rlv3841 determined by INSeq (13, 31). Genes that were experimentally classified as essential or defective were considered essential in our analysis. The threshold for a gene to be considered essential *in silico* was set to 50% of the wild-type growth rate because all mutant strains are grown in a single culture for an INSeq experiment, and a slower growth rate will therefore decrease the abundance of a mutant even if the gene carrying the insertion is not absolutely essential. Growth rates were determined by allowing unlimited uptake of UMS media components and ammonia while either glucose or succinate was provided as the sole carbon source. Uptake rates of glucose and succinate were constrained to the experimentally determined values of 1.710 mmol h$^{-1}$ per g cellular dry weight and 3.224 mmol h$^{-1}$ per g cellular dry weight, respectively (41).

For comparison with $^{13}$C labeling data, boundary conditions were set to allow for unlimited uptake of UMS media components. The succinate uptake rate was constrained to 1 flux unit and flux balance analysis was performed maximizing the biomass objective function. In addition, loopless flux variability analysis was performed where the objective fraction was set to 95% of the optimum value.

**Data integration for model contextualization.** The Python implementation of RIPTiDe (https://github.com/mjenior/riptide) (44) was used to generate condition-specific models of iCS1224. Max fit RIPTiDe was run for objective flux fractions between 0.5 and 0.95 with 0.05 increments, and the context-specific models with the highest correlation between flux values and transcriptome data were used in further analyses.

In addition to the nutrient availability determined based on experimental data, trace elements and vitamins required for flux through the objective function were added to the *in silico* representation of each environment. If an exchange and transport reaction for a compound already existed in the model, the lower bound of the exchange reaction was set to −1,000. If a compound was only present as an intracellular metabolite, a sink reaction for this metabolite with lower bound set to −1,000 and upper bound set to 0 was added. This was done to avoid erroneous exclusion of metabolites which are present in the environment, but for which transporters have not been identified. Some cofactors and central intermediates of carbon metabolism, such as glyceraldehyde 3-phsophate, were excluded from environmental representations because their uptake would result in unspecific predictions for metabolic pathway activity.

**Data integration for rhizosphere model.** For the rhizosphere model, compounds that have been detected in pea root exudates (10, 12, 51, 52) and that could be matched to model metabolites were specified with unlimited availability (Table S2). Flux through the biomass reaction as described in the previous section was set as the objective function and, in addition, a lower bound of one flux unit was set for demand reactions for Nod factor, lipopolysaccharides, and exopolysaccharides, because these compounds are known to be produced as part of the root colonization process (4). RPKM values for RNA-Seq data obtained from Rlv3841 in the pea rhizosphere 7 days postinoculation were provided as an input, and all genes that are present in the model and were classified as essential or defective in the rhizosphere (13) were specified as model tasks to prevent removal of the associated reactions during the pruning process.

The reporter metabolite algorithm was implemented as previously described (63) using the `reporterMetabolites` function from the RAVEN Toolbox 2.0 (85).

**Data integration for nodule bacteria model.** For nodule bacteria, all metabolites that were detected by rhizobial biosensors in pea nodules (12) were allowed to be taken up without limitation, as well as all amino acids and metabolites whose abundance was at least 10-fold higher in the nodule cytosol compared to root exudates (Table S3). A lower bound of one flux unit was set for lipopolysaccharide and exopolysaccharide demand reactions and biomass production was used as the objective function. RPKM values for dRNA-Seq data obtained from the tip of pea nodules were provided as an input, and all genes that were classified as essential or defective for nodule bacteria (13) were specified as model tasks to prevent removal of the associated reactions during the pruning process.

**Data integration for bacteroids.** Similar to the nodule bacteria model, metabolites detected in nodules by rhizobial biosensors, and all amino acids were made available to the bacteroid model (Table S4). However, fructose and sucrose were not included since they are known to be poorly oxidized by

bacteroids (97). Inclusion of the metabolites increased in the nodule cytosol compared with root exudates led to a decrease in correlation of flux predictions and gene expression data, and those metabolites were therefore omitted from the nutrients available to bacteroids. A lower bound of 1 flux unit was set for the synthesis of fatty acids and proteins and flux through the nitrogenase reaction was used as the objective function. RPKM values for dRNA-Seq data obtained from the middle of pea nodules were provided as an input, and all genes associated with proteins significantly upregulated in bacteroids compared to free-living cells (23) were specified as model tasks to prevent removal of the associated reactions during the pruning process.

**Phenotype MicroArray™ analysis.** Carbon source utilization of Rlv3841 was assessed using the phenotype microarray technology (Biolog, Hayward, USA). A liquid culture of Rlv3841 was grown at 28°C in UMS media supplemented with 10 mM glucose, 10 mM ammonium chloride, and a vitamin solution as previously described (31). Cells were spun down and washed three times in UMS without addition of a carbon or nitrogen source. The optical density at 600 nM was then adjusted to 0.1 with UMS supplemented with 10 mM ammonium chloride and vitamins, and 100 $\mu$L of cell suspension were added to each well of the phenotype microarray plate. After overnight incubation without shaking at 28°C, 10 $\mu$L of a 0.1% (wt/vol) stock solution of 2,3,5-triphenyltetrazolium chloride in water were added to each well. Plates were then incubated in an Omega FluoStar plate reader with double orbital shaking at 500 rpm and the absorbance at 505 nm was measured every 15 min. Absorbance values were analyzed using the DuctApe software (98), and all carbon sources with an activity value higher than the water control were considered to support growth. For activity values close to the growth threshold, curves were manually inspected, and literature searches were performed to determine if the carbon source supports growth of *R. leguminosarum*. The full DuctApe output for the phenotype microarray analysis is available in Data set S1.

**Metabolomics data.** Metabolomics data were obtained in a previous study (43), where only values for metabolites relevant to the investigated metabolic pathways were published. The full metabolomics data set is included as Data set S2.

**Sample preparation for RNA-Seq of rhizosphere bacteria.** For total RNA extraction from Rlv3841 in the pea rhizosphere, *Pisum sativum* cv. Avola seeds were surface sterilized and sown in sterilized boiling tubes with fine vermiculite and nitrogen-free rooting solution. Pea seeds were grown in the dark for 3 days and then transferred to a controlled environment room, where they were grown at 25°C with a 16:8-h photoperiod for another 4 days. On day 7, 1 mL ($10^8$ CFU/mL) of washed Rlv3841 cells was added near the root. At 7 days postinoculation, rhizobial cells were harvested from the rhizosphere as previously described (10). RNA was extracted for three biological replicates where the total RNA extracted from the pea rhizosphere of 16 boiling tubes was pooled for each replicate. Quality and quantity of the total RNA was assessed using Experion StdSens (Standard Sensitivity) and HighSens (High Sensitivity) analysis kits. Total RNA (3 $\mu$g per sample) was treated with the TURBO DNA-free kit (Invitrogen AM1907) as previously described (10). Depletion of genomic DNA was confirmed by performing a Qubit fluorometer double-stranded DNA broad range assay. Finally, the rRNA was depleted from the RNA sample using the Illumina Ribo-Zero rRNA removal kit, Gram-negative (MRZGN126) according to the manufacturer's instructions. The rRNA-depleted mRNA was purified using the ZymoResearch RNA Clean & Concentrator 50. mRNA samples were used to generate barcoded cDNA libraries for multiplexing during sequencing using the Ion Total RNA-Seq kit v2 (Thermo Fisher Scientific). Each barcoded cDNA library was quantified using the Agilent Bioanalyzer High Sensitivity DNA kit and diluted to a final concentration of 70 pM. Equal volumes of the diluted cDNA libraries were pooled before loading on the IonChef for template preparation and chip loading. Finally, the chips were sequenced in an Ion Proton Semiconductor based sequencing platform (Thermo Fisher Scientific). The full data set is available on the NCBI SRA database, BioProject number PRJNA748006.

**Data analysis for RNA-Seq of rhizosphere bacteria.** RNA-Seq data was de-multiplexed based on valid barcodes and data for each library was downloaded in fastq format. The overall quality of the sequencing and the data was assessed based on the Torrent Browser suite sequencing run report summary. Data from each library was assessed using FastQC (Babraham Institute; https://www.bioinformatics .babraham.ac.uk/projects/fastqc/) and any remaining adapters and low-quality reads were filtered using cutadapt (99). The data for each library was mapped against the Rlv3841 genome using EDGE-pro (100) developed specifically for bacterial RNA-Seq data. EDGE-pro uses Bowtie2 to map the reads to the genome and calculates the frequencies per nucleotide. EDGE-pro calculates the number of reads and RPKM value for each gene feature in the genome including mRNA, rRNA, and tRNA. The mapped reads from each library were visualized with the Integrative Genomics Viewer (101) for further analysis.

**dRNA-Seq data for nodule bacteria and bacteroids.** The dRNA-Seq data used for creation of the nodule bacteria and the bacteroid model were described previously (67) and are available on the NCBI SRA database, BioProject number PRJNA748006.

**Data availability.** All data needed to evaluate the conclusions in this article are present in the article and/or the supplementary materials. RNA-Seq data for Rlv3841 in the pea rhizosphere are available on the NCBI SRA database, BioProject number PRJNA748006. All code and metabolic models are available on Github (https://github.com/CarolinSchulte/Rlv3841-lifestyles).

## SUPPLEMENTAL MATERIAL

Supplemental material is available online only.
**DATA SET S1,** XLSX file, 0.03 MB.
**DATA SET S2,** XLSX file, 0.2 MB.

**FIG S1**, EPS file, 1.3 MB.
**FIG S2**, EPS file, 1.2 MB.
**FIG S3**, EPS file, 1 MB.
**TABLE S1**, DOCX file, 0.02 MB.
**TABLE S2**, DOCX file, 0.02 MB.
**TABLE S3**, DOCX file, 0.01 MB.
**TABLE S4**, DOCX file, 0.01 MB.
**TABLE S5**, DOCX file, 0.01 MB.

## ACKNOWLEDGMENTS

This work was supported by funding from the Biotechnology and Biological Sciences Research Council (BBSRC) (Grant numbers BB/M011224/1, BB/R017859/1, BB/T001801/1, and BB/T006722/1). C.C.M.S. is supported by the Clarendon Fund (Oxford University) and the Keble College De Breyne Scholarship. A.P. was funded in part by the Engineering and Physical Sciences Research Council (EPSRC) (Grant number EP/M002454/1).

Conceptualization: C.C.M.S., A.P., P.S.P.; Methodology: C.C.M.S., A.P., P.S.P.; Formal analysis: C.C.M.S., V.K.R.; Investigation: C.C.M.S., V.K.R., A.P., P.S.P.; Visualization: C.C.M.S.; Supervision: A.P., P.S.P.; Writing—original draft: C.C.M.S.; Writing—review & editing: C.C.M.S., V.K.R., A.P., P.S.P.

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
