## [Reviewer comments · mSystems]

Genome-scale metabolic modelling of lifestyle changes in *Rhizobium leguminosarum*

Carolin Schulte, Vinoy Ramachandran, Antonis Papachristodoulou, and Philip Poole

Corresponding Author(s): Philip Poole, Oxford University

Review Timeline:

Submission Date:	July 27, 2021
Editorial Decision:	September 24, 2021
Revision Received:	November 24, 2021
Accepted:	December 20, 2021

Editor: E. Sogin

Reviewer(s): Disclosure of reviewer identity is with reference to reviewer comments included in decision letter(s). The following individuals involved in review of your submission have agreed to reveal their identity: Marco Fondi (Reviewer #2)

Transaction Report:

DOI: <https://doi.org/10.1128/mSystems.00975-21>

September 24, 2021

Prof. Philip Poole
Oxford University
Department of Plant Sciences
Oxford Oxford OX1 3RB
United Kingdom

Re: mSystems00975-21 (Genome-scale metabolic modelling of lifestyle changes in *Rhizobium leguminosarum*)

Dear Prof. Philip Poole:

Thank you for submitting your manuscript to mSystems. We have completed our review and I am pleased to inform you that, in principle, we expect to accept it for publication in mSystems. However, acceptance will not be final until you have adequately addressed the reviewer comments.

Prior to resubmission, please reduced the number of supplemental items to below 10 as per to the requirements of mSystems.

Preparing Revision Guidelines

Sincerely,

E. Sogin

Editor, mSystems

Journals Department
Reviewer comments:

Reviewer #2 (Comments for the Author):

In this work, Schulte et al. has assembled and validated a metabolic reconstruction for the plant symbiont *Rhizobium leguminosarum*. The reconstruction has been integrated with different -omics datasets (including transcriptomics, metabolomics, gene essentiality data)

They have then used the model to study the metabolic reprogramming occurring in these cells during all the different stages of the rhizobium-legume symbioses. The authors have identified specific metabolites and reactions that appear to play a role in the adaptation to each specific lifestyle of the bacterium and that deserve further attention and/or experimental validation. The manuscript follows the same general path of other manuscripts describing the reconstruction of plant symbionts (e.g. *S. meliloti* 10.1038/s41467-020-16484-2 or the same authors in DOI: 10.1126/sciadv.abh2433) both in the kind of analyses performed, in the tools used for this purpose and (partially) in the biological outcomes.

The authors have made the reconstruction available as Supplementary Material and I praise the authors for this. I have downloaded it and used it for some basic simulations and the format is compatible with the most common modelling tools.

The authors have done a very good job in putting together an apparently reliable reconstruction and in using it for deriving valuable biological insights. The reconstruction represents a solid platform for future systems biology studies on plant-bacteria interactions.

Here is the list of my major concerns.

- L. 162. I would like to see these points addressed: 1. How did the authors simulated growth on complex medium? It is known that the definition of nutrients uptake may have profound effects on the resulting simulations so the definition of the actual active EX(CHANGE) reactions set is crucial here to assess the reliability of these simulations. Please clearly state how the nutritional requirements of the model were set to reproduce the experimental conditions. 2. Some indications on how many essential genes have been compared here would be useful. How many genes were predicted to be essential during INseq experiments and/or during model simulations? 3. It would also be interesting to see the genes/reactions in which experiments and models do not agree. These cases may represent an occasion for improving the reconstruction and/or for pinpointing interesting biological facts.

- L.177. While I agree that the validity of the reconstruction has been (qualitatively) tested I am a bit perplexed by the absence of a quantitative evaluation. Usually, growth rates predicted by genome-scale reconstructions are compared against those obtained experimentally, in the same nutritional conditions (on minimal medium). The authors should check whether such experimental data are available for *R. leguminosarum* and, in case, include this benchmark in the manuscript.

- L. 201. "exchange reactions were added to the model with reaction bounds set to allow for unlimited uptake.". I am not entirely convinced by this approach. Un-constraining the boundaries of exchange reactions usually results in unrealistic fluxes across (at least some) of the reactions of the model. It is true that the authors used expression data to constraint the activity of downstream reactions but I encourage the use of more appropriate (lower, I would say) nutrients uptake values and double check the simulation outcomes.

- L. 208-210. Again, details should be added here on the values of "additional positive lower bounds" included in the model. Were these boundaries set to simulate the (forced) secretion of these compounds by the model? Please clarify.

- L. 244. The text is a bit misleading here. Does "upregulated" refer to a gene or a reaction in the model once expression data has been mapped?

- A genome-scale reconstruction of *R. etli* is available since 2007. Although much smaller than the one presented here, did the authors use it as a source of GPR and/or reactions for the reconstruction of *R. leguminosarum* model? In any case, I think that the presence of this (and others?) *Rhizobium* reconstructions should be acknowledged.

L. 616-617. RNA and protein content for biomass reaction have been estimated on the basis of sequence and literature information. However, other methods exist to estimate the stoichiometric coefficients of these macromolecules in the biomass reaction (e.g. 10.1371/journal.pcbi.1006971). I would suggest adopting this approach or a similar one (based on experimental data, e.g. transcriptomics) for a more reliable estimation of these coefficients in the reconstruction. Are (and, if yes, to what extent) modelling results affected?

L. 438. Metabolism of bacteroids. I am concerned by the novelty of this entire section. In a previous work (DOI: 10.1126/sciadv.abh2433), the authors have already reconstructed a model specific for nitrogen-fixing bacteroids for *R. leguminosarum* bv. *viciae* 3841. I don't understand the point in reconstructing a novel bacteroid model. The only reason would be

some differences in two models or a different way of assembling them. I would like to see this all discussed in the manuscript and the current bacteroid reconstruction compared with the previous one in terms of n. of reactions, genes, and so on.

Reviewer #3 (Comments for the Author):

The manuscript by Schulte et al. reports the construction and validation of a genome-scale metabolic model for the plant nitrogen fixing symbiont *Rhizobium leguminosarum* bv. *viciae* 3841. The model has been based on an initial draft reconstruction from already available models in relative rhizobial species, further refined by published data on essential genes (InSeq) C-13 flux and metabolomics, and original data on substrate utilization and RNA-Seq in the pea rhizosphere. Data obtained are robust and the model is well in agreement with experiment data, providing a valuable resource for researchers in the field of plant-microbe interaction and in nitrogen-fixing symbiosis.

Prior to publication few points deserve further attention and/or improvement

1. The paragraph on reporter metabolites (Fig. 5), based on already published microarray data is unclear. How are reported metabolites been obtained? The method section need further explanation to understand how single metabolites have been identified and understand the biological significance of the data obtained.
2. I appreciate conciseness in the Discussion section. However, a comparison with results obtained from the other similar systems indicated in the introduction section (e.g. *B. japonicum*, *S. meliloti*) would increase the interest of readers toward the present data and may highlight common vs. specific features of the transition from rhizosphere to nodule symbiosis in different species.

Response to reviewers

All line numbers refer to the marked up manuscript

Reviewer #2 (Comments for the Author):

In this work, Schulte et al. has assembled and validated a metabolic reconstruction for the plant symbiont *Rhizobium leguminosarum*. The reconstruction has been integrated with different -omics datasets (including transcriptomics, metabolomics, gene essentiality data)

They have then used the model to study the metabolic reprogramming occurring in these cells during all the different stages of the rhizobium-legume symbioses. The authors have identified specific metabolites and reactions that appear to play a role in the adaptation to each specific lifestyle of the bacterium and that deserve further attention and/or experimental validation. The manuscript follows the same general path of other manuscripts describing the reconstruction of plant symbionts (e.g. *S. meliloti* 10.1038/s41467-020-16484-2 or the same authors in DOI: 10.1126/sciadv.abh2433) both in the kind of analyses performed, in the tools used for this purpose and (partially) in the biological outcomes.

The authors have made the reconstruction available as Supplementary Material and I praise the authors for this. I have downloaded it and used it for some basic simulations and the format is compatible with the most common modelling tools.

The authors have done a very good job in putting together an apparently reliable reconstruction and in using it for deriving valuable biological insights. The reconstruction represents a solid platform for future systems biology studies on plant-bacteria interactions.

We appreciate the reviewer's effort to ensure the reliability of our model and thank the reviewer for their positive feedback.

Here is the list of my major concerns.

- L. 162. I would like to see these points addressed: 1. How did the authors simulated growth on complex medium? It is known that the definition of nutrients uptake may have profound effects on the resulting simulations so the definition of the actual active EX(change) reactions set is crucial here to assess the reliability of these simulations. Please clearly state how the nutritional requirements of the model were set to reproduce the experimental conditions. 2. Some indications on how many essential genes have been compared here would be useful. How many genes were predicted to be essential during INseq experiments and/or during model simulations? 3. It would also be interesting to see the genes/reactions in which experiments and models do not agree. These cases may represent an occasion for improving the reconstruction and/or for pinpointing interesting biological facts.

1. We did not perform simulations on complex media, but rather used gene essentiality data generated for growth in complex media to obtain a comprehensive list of genes that are essential for free-living growth. The reason for this approach is that there is some variability in gene essentiality assignments obtained by insertion sequencing experiments (as described in Blazier and Papin, 2019). To clarify that the gene essentiality predictions by our model do not include growth on complex media, we have modified the corresponding sentence to read "Predictions by iCS1224 for gene essentiality during growth in minimal media" (line 163).
2. The numbers indicating predictive performance for essential and non-essential genes are shown in Fig. 2B.
3. This information has been added to the Github repository.

- L.177. While I agree that the validity of the reconstruction has been (qualitatively) tested I am a bit perplexed by the absence of a quantitative evaluation. Usually, growth rates predicted by genome-scale reconstructions are compared against those obtained experimentally, in the same nutritional conditions (on minimal medium). The authors should check whether such experimental data are available for *R. leguminosarum* and, in case, include this benchmark in the manuscript.

A comparison of predicted and experimentally determined growth rates for the carbon sources glucose and succinate has been added to the Results section (line 173-181). We would like to point out that quantitative validation of fluxes through central carbon metabolism has been performed by comparing model predictions with ¹³C-MFA data (Fig. 2C).

- L. 201. "exchange reactions were added to the model with reaction bounds set to allow for unlimited uptake.". I am not entirely convinced by this approach. Un-constraining the boundaries of exchange reactions usually results in unrealistic fluxes across (at least some) of the reactions of the model. It is true that the authors used expression data to constraint the activity of downstream reactions but I encourage the use of more appropriate (lower, I would say) nutrients uptake values and double check the simulation outcomes.

We thank the reviewer for this comment. We agree that our approach does not allow for quantitative evaluation of metabolic fluxes. However, since uptake fluxes for the complex set of metabolites in the rhizosphere cannot be experimentally determined, we believe that constraining uptake rates in the model to arbitrary values would bias our analysis. We therefore decided to perform a qualitative assessment of metabolic pathways predicted to be active in the rhizosphere solely based on metabolome and transcriptome data. This aligns with the recommendation of the authors of the RPTiDe algorithm (Jenior *et al.*, 2020), who suggest leaving exchange reaction bounds unconstrained during model contextualization, in particular when environmental conditions and/or uptake fluxes cannot be accurately defined.

- L. 208-210. Again, details should be added here on the values of "additional positive lower bounds" included in the model. Were these boundaries set to simulate the (forced) secretion of these compounds by the model? Please clarify.

As the reviewer correctly assumed, the positive lower bounds were added to force the production of polysaccharides and Nod factors, which is known to occur in the rhizosphere, and prevent removal of the corresponding reactions from the contextualized model. The sentence has been modified to clarify this (line 228-229).

- L. 244. The text is a bit misleading here. Does "upregulated" refer to a gene or a reaction in the model once expression data has been mapped?

The corresponding sentence has been changed to "The **gene encoding the solute-binding protein** [...] was upregulated" (line 264).

- A genome-scale reconstruction of *R. etli* is available since 2007. Although much smaller than the one presented here, did the authors use it as a source of GPR and/or reactions for the reconstruction of *R. leguminosarum* model? In any case, I think that the presence of this (and others?) *Rhizobium* reconstructions should be acknowledged.

We apologize for unintentionally omitting this reconstruction from the list of references, the reference has been added to the introduction (line 91), which also includes references to all other models for rhizobia that we are aware of. The *R. etli* model has not been used in our model reconstruction process.

L. 616-617. RNA and protein content for biomass reaction have been estimated on the basis of sequence and literature information. However, other methods exist to estimate the stoichiometric coefficients of these macromolecules in the biomass reaction (e.g. 10.1371/journal.pcbi.1006971). I would suggest adopting this approach or a similar one (based on experimental data, e.g. transcriptomics) for a more reliable estimation of these coefficients in the reconstruction. Are (and, if yes, to what extent) modelling results affected?

We thank the reviewer for this suggestion. While we agree that using a biomass function specific to each environment would be desirable, the central piece of experimental data required for the approach suggested by the reviewer would be the molecular weight fraction of the main cellular components for each environment. Since experimental data for the biomass composition of rhizosphere bacteria and nodule bacteria are not available, we decided to use the more general biomass objective function for free-living growth throughout and aimed to provide some flexibility by including sink reactions for carbon polymers.

L. 438. Metabolism of bacteroids. I am concerned by the novelty of this entire section. In a previous work (DOI: 10.1126/sciadv.abh2433), the authors have already reconstructed a model specific for nitrogen-fixing bacteroids for *R. leguminosarum* bv. *viciae* 3841. I don't understand the point in reconstructing a novel bacteroid model. The only reason would be some differences in two models or a different way of assembling them. I would like to see this all discussed in the manuscript and the current bacteroid reconstruction compared with the previous one in terms of n. of reactions, genes, and so on.

We thank the reviewer for this comment. To distinguish the bacteroid model presented in this work from our previously published model, we have added a description of differences in the model reconstruction process as well as a comparison of reaction content (lines 479-482 and 488-498). While we are aware that there is some overlap with our previous work, we consider the inclusion of a bacteroid model essential for the completeness of describing the entire "lifestyle transition process" of Rlv3841. In our recently published work, the bacteroid model was obtained through a "bottom-up" approach to focus on a minimalistic core model of central carbon and nitrogen metabolism. In contrast, in this work we obtain the bacteroid model starting from a full genome-scale model, which leaves more flexibility for the inclusion of accessory metabolism or metabolic pathways that have so far not been described to be active in bacteroids. We believe that the novel predictions that were generated regarding uptake of glycolate and xylose as well as histidine synthesis by bacteroids are important and informative for future experimental studies and should therefore be included in this work.

Reviewer #3 (Comments for the Author):

The manuscript by Schulte et al. reports the construction and validation of a genome-scale metabolic model for the plant nitrogen fixing symbiont *Rhizobium leguminosarum* bv. *viciae* 3841. The model has been based on an initial draft reconstruction from already available models in relative rhizobial species, further refined by published data on essential genes (InSeq) C-13 flux and metabolomics, and original data on substrate utilization and RNA-Seq in the pea rhizosphere. Data obtained are robust and the model is well in agreement with experiment data, providing a valuable resource for researchers in the field of plant-microbe interaction and in nitrogen-fixing symbiosis.

We thank the reviewer for their positive feedback on our work.

Prior to publication few points deserve further attention and/or improvement

1. The paragraph on reporter metabolites (Fig. 5), based on already published microarray data is unclear. How are reported metabolites been obtained? The method section need further explanation to understand how single metabolites have been identified and understand the biological significance of the data obtained.

We thank the reviewer for their comment. We have refrained from a more detailed explanation of the reporter metabolite algorithm due to space constraints and because the algorithm is described in detail in the original publication (reference (63)) and has been used in various metabolic modelling studies (e.g., Mardinoglu *et al.* 2014, Wang *et al.* 2021). We agree that a description of the implementation is important for reproducibility of our work and we have therefore added the following sentence to the methods section: “The reporter metabolite algorithm was implemented as previously described (63) using the `reporterMetabolites` function from the RAVEN Toolbox 2.0 (85).” (line 776-777)

2. I appreciate conciseness in the Discussion section. However, a comparison with results obtained from the other similar systems indicated in the introduction section (e.g. *B. japonicum*, *S. meliloti*) would increase the interest of readers toward the present data and may highlight common vs. specific features of the transition from rhizosphere to nodule symbiosis in different species.

We thank the reviewer for their suggestion to improve the relevance of our work for a broader audience. Relevant comparisons with models for other rhizobial species have been added at several places in the discussion (lines 588-591, 612-617, 623-626). We would further like to point out that comparisons to previous modelling studies are included throughout the Results section to facilitate contextualization of the predictions from our model.

December 20, 2021

Prof. Philip Poole
Oxford University
Department of Plant Sciences
Oxford Oxford OX1 3RB
United Kingdom

Re: mSystems00975-21R1 (Genome-scale metabolic modelling of lifestyle changes in *Rhizobium leguminosarum*)

Dear Prof. Philip Poole:

Your manuscript has been accepted, and I am forwarding it to the ASM Journals Department for publication. For your reference, ASM Journals' address is given below. Before it can be scheduled for publication, your manuscript will be checked by the mSystems senior production editor, Ellie Ghatineh, to make sure that all elements meet the technical requirements for publication. She will contact you if anything needs to be revised before copyediting and production can begin. Otherwise, you will be notified when your proofs are ready to be viewed.

Publication Fees:

We recognize that the video files can become quite large, and so to avoid quality loss ASM suggests sending the video file via <https://www.wetransfer.com/>. When you have a final version of the video and the still ready to share, please send it to mssystemsjournal@msubmit.net.

Sincerely,

E. Sogin
Editor, mSystems

Journals Department
Table S5: Accept

Fig. S3: Accept

Table S1: Accept

Table S2: Accept

Data S1: Accept

Table S3: Accept

Table S4: Accept

Data S2: Accept

Fig. S2: Accept

Fig. S1: Accept